# Ketogenic Diet in the Treatment of Gliomas and Glioblastomas

**DOI:** 10.3390/nu14183851

**Published:** 2022-09-17

**Authors:** Simone Dal Bello, Francesca Valdemarin, Deborah Martinuzzi, Francesca Filippi, Gian Luigi Gigli, Mariarosaria Valente

**Affiliations:** 1Clinical Neurology Unit, Santa Maria della Misericordia University Hospital, 33100 Udine, Italy; 2Department of Medical Area, University of Udine, 33100 Udine, Italy

**Keywords:** ketogenic diet, ketosis, ketones, cancer, tumor, glioma, glioblastoma, Warburg effect

## Abstract

In recent years, scientific interest in the use of the ketogenic diet (KD) as a complementary approach to the standard cancer therapy has grown, in particular against those of the central nervous system (CNS). In metabolic terms, there are the following differences between healthy and neoplastic cells: neoplastic cells divert their metabolism to anaerobic glycolysis (Warburg effect), they alter the normal mitochondrial functioning, and they use mainly certain amino acids for their own metabolic needs, to gain an advantage over healthy cells and to lead to a pro-oncogenetic effect. Several works in literature speculate which are the molecular targets of KD used against cancer. The following different mechanisms of action will be explored in this review: metabolic, inflammatory, oncogenic and oncosuppressive, ROS, and epigenetic modulation. Preclinical and clinical studies on the use of KD in CNS tumors have also increased in recent years. An interesting hypothesis emerged from the studies about the possible use of a ketogenic diet as a combination therapy along with chemotherapy (CT) and radiotherapy (RT) for the treatment of cancer. Currently, however, clinical data are still very limited but encouraging, so we need further studies to definitively validate or disprove the role of KD in fighting against cancer.

## 1. Introduction

Brain and other central nervous system (CNS) tumors, while rare, cause significant morbidity and mortality across all ages [1]. Gliomas are the primary brain tumors of the central nervous system. The prognosis, especially for high-grade gliomas, is dismal; the treatment of these tumors represents an unmet need in medicine [2]. In recent years, scientific interest in the use of the ketogenic diet (KD) as a complementary approach to standard cancer therapy has grown, in particular against those of the CNS [3]. This diet has been used to treat drug-resistant childhood epilepsy since the 1920s [4]. Currently, literature-based evidence suggests an important role of KD in the therapy of both neurological and non-neurological diseases, in particular, the following: epilepsy, migraine headache, Alzheimer’s disease, motor neuron disease, autism, multiple sclerosis, Parkinson’s disease, diabetes, obesity, cancer, acne, polycystic ovary syndrome, glucose transporter type 1 (GLUT-1) and pyruvate dehydrogenase complex (PDH) deficiency [5,6,7]. (Figure 1).

The ketogenic diet includes any intervention that induces the organism to produce ketones (β-hydroxybutyrate, acetoacetate, and acetone) as the only alternative form of energy for the body to glucose; ketones are formed from ingesting fatty acids (while using ketogenic high-fat diets) or producing endogenous fatty acids (as during fasting and in hypoglucidic-hypolipidic-normoprotein diets) [8]. Indeed, KD production is stimulated by the glucose deficit and by the reduction of the insulin/glucagon ratio as follows: Due to maintaining blood sugar levels constant to avoid hypoglycemic coma, hepatocytes need to use all the oxalate available for gluconeogenesis, causing the block of the Krebs cycle’s first step; even in the absence of available oxaloacetate, acetyl-CoA will continue to be abundantly produced through the beta-oxidation of fatty acids, thus leading to its cytoplasmic accumulation. The accumulated acetyl-CoA will then be directed towards the production of acetoacetate, which will then spontaneously be converted to acetone (a volatile compound that is eliminated with exhalation) and 3-β-hydroxybutyrate (the predominant ketone body, about 70% of the total, used as an energy source) by 3-β-hydroxybutyrate dehydrogenase. Once produced, KD will be released into the bloodstream reaching the peripheral cells where, after entering into the cytosol, they will be reconverted to acetyl-CoA. Now acetyl-CoA is allowed to enter into the Krebs cycle; so, there is the availability of oxaloacetate inside peripheral tissues that are not able to do gluconeogenesis [9]. (Figure 2) It was recently discovered that although hepatocytes are the main producers of ketone bodies, astrocytes, T cells, intestinal stem cells LGR5+, and cells of the renal epithelium are also able to produce these molecules, although in much smaller quantities [10,11].

There are the following multiple ketogenic dietary interventions used in clinical practice:The Classic Ketogenic Diet (CKD), in which the ratio between fats and non-fats (carbohydrates + proteins) must be calculated; generally, this ratio is 3:1 or 4:1 (i.e., the intake in grams of fats is three or four times that of non-fats). This protocol is characterized by the higher content of fats compared to the protein portion (slightly reduced or normal) and carbohydrates (greatly reduced) [12];Modified Atkins Diet (MAD): the Atkins diet was created in the 1970s as a means to combat obesity. This diet allows more protein, and calories are not closely monitored. You can also start it without a fast. Calorie intake is 60% from fat, 30% from protein, and 10% from carbohydrates [12];Medium-Chain Triglyceride (MCT): Dr. Peter Huttenlocher and his group replaced some of the long-chain fat in the classic ketogenic diet—that is, fat from foods such as butter, oils, cream, and mayonnaise—with an alternative fat source with a shorter carbon chain length. This medium-chain fat, otherwise known as medium-chain triglyceride (MCT), is absorbed more efficiently than long-chain fat, is carried directly to the liver in the portal blood, and does not require carnitine to facilitate transport into cell mitochondria for oxidation. Because of these metabolic effects, MCT will yield more ketones per kilocalorie of energy than its long-chain counterparts. This increased ketogenic potential means less total fat is needed in the MCT diet. Whereas the classical 4:1 ratio ketogenic diet provides 90% of energy from fat, the MCT ketogenic diet typically provides 70% to 75% of energy from fat (both MCT and long-chain), allowing more protein and carbohydrate food to be included [5];The Very Low-Calorie Ketogenic Diet (VLCKD), an extremely restrictive nutritional protocol (600–800 kcal), limited in time (up to 12 weeks), characterized by a minimum protein content (≥75 g/day), a very limited carbohydrate content (30–50 g/day), a fixed amount of fat (20 g/day, mainly from olive oil and omega-3), and micronutrients to meet the Dietary Reference Intake (DRI), in accordance with the European Food Safety Authority (EFSA) [13].

## 2. Metabolic Differences between Healthy and Cancer Cells

In metabolic terms, there are differences between healthy and neoplastic cells in the production of the ATP required for cellular sustainment. Healthy cells, under standard conditions, use aerobic metabolic pathways that, starting from various substrates, converge on the Krebs cycle for energy synthesis. The main substrates used in this context are carbohydrates as follows: they are catabolized in glycolysis, a cytoplasmic anaerobic pathway ubiquitous in living beings, generating ATP (5.6% of available energy), NADH, and pyruvate. The pyruvate produced may be converted to acetyl-CoA and enter the Krebs cycle (under aerobic conditions) or be converted to lactate (under anaerobic conditions), which is the final product of anaerobic glycolysis [14,15].

In contrast, cancer cells appear to develop a predominantly anaerobic metabolism, with overexpression and overutilization of anaerobic glycolysis as the main cellular energy source. This metabolic variation is named the “Warburg effect” after the first scientist who hypothesized this theory (1926) [16]. (Figure 3) In addition, in later years, it has been hypothesized that there is a bidirectional relationship between the mutational rate typical of tumors and the metabolic change of tumors, as follows: mutation of oncogenes and oncosuppressors can induce a change in metabolism, and this itself seems to be able to make the tumor more prone to acquiring mutations. Reinforcing this hypothesis is also the evidence that increased lactate levels (and the consequent reduction in pH) may promote metastatic processes; in addition, cellular hypoxia would go on to promote the overexpression of HIF-1, a transcriptional factor that stimulates the production of VEGF, EPO, glycolytic enzymes, and GLUT [17]. The end result of the metabolic change is a complete subversion of cellular metabolic mechanisms, with the modification of normal mitochondrial functioning rerouted to the production of ROS, carbon skeletons, and other molecules useful for tumor growth and proliferation; glycolysis and other cellular metabolic pathways are also disrupted in order to stimulate uncontrolled replication, resistance to apoptosis, local and distant invasion, and undifferentiation [14]. 

Regarding the metabolic shift of the cancer cell, it must be remembered that anaerobic glycolysis allows the production of only 2 ATP molecules for each glucose molecule, compared with a total of 36 ATP molecules produced by the aerobic pathway. One possible reason for this energetically disadvantageous choice may be the remarkable speed of this anaerobic pathway compared with the more cumbersome aerobic process; this metabolic modification may be useful because it gives the cancer cell an important advantage in terms of the speed of acquisition of available extracellular resources at the expense of healthy cells [15]. The result of this process is that the neoplastic cell, compared with the healthy cell, consumes at least 10 times more glucose and produces at least 100 times more lactate [18]. An additional advantage related to these types of cancer cells lies in the possibility of synthesis of various molecules essential for cell growth and proliferation. Starting from anaerobic glycolysis, indeed, it is possible to synthesize nucleotides, fatty acids, and amino acids, and in addition, it is possible to use some glycolytic intermediates such as dihydroxyacetone phosphate for the biosynthesis of phospholipids and triglycerides. In addition, glucose 6-phosphate can be redirected to the pentose phosphate shunt to synthesize ribose-5-phosphate for nucleotide and NADPH synthesis [14].

Metabolic changes typical of the tumor environment are not limited to glycolysis but they can also affect mitochondrial function. Indeed, tumor cells can induce a constitutive and continuous activation of the Krebs cycle by using not pyruvate but alpha-ketoglutarate, derived from glutaminolysis, as its substrate; in this way, the tumor can also use Krebs cycle intermediates for the synthesis of indispensable molecules (nucleic acids, proteins, and lipids) [14].

The neoplastic cell, in addition to altering glucose metabolism, also alters amino acid synthesis and utilization; indeed, it has been observed that in tumors there is an increased intake and utilization of the following:Glutamine: a crucial amino acid used as a mitochondrial energy source for the synthesis of various molecules and for its antioxidant power through the synthesis of glutathione. Glutamate produced from glutamine can also be used as a starting substrate for the synthesis of other nonessential amino acids such as aspartate, alanine, arginine, and proline. In addition, by reductive carboxylation, glutamine is converted to citrate and is furthermore used for lipid synthesis. Glutamine is also a source of nitrogen for the biosynthesis of glycosylated molecules and nucleotides [14,19];Serine: a monocarbon unit donor amino acid useful for folate synthesis. The biosynthesis pathway of this amino acid also contributes to the regulation of glycolysis and glutaminolysis pathways typical of cancer [14,20];Leucine: essential amino acid crucial for protein synthesis [21];Aspartate: amino acid fundamental for cellular protein synthesis and nucleotide biosynthesis, which is crucial especially in cells with a high proliferative index; the increase in its synthesis appears to be related to mitochondrial dysregulation [14].

Recently, the hypothesis of a “reverse Warburg effect” has emerged. This theory assumes that the metabolic changes found within the tumor are not limited to the pathological tissue but extend to and involve neighboring healthy cells as well. In this context, the tumor is described as a “parasite” that induces catabolic processes in adjacent healthy cells by absorbing their produced metabolites. Indeed, it has been shown that lactic acid, produced through glycolysis both at the tumor level and especially by peritumoral fibroblasts, could be used by neoplastic cells for citric acid production (an energy substrate), and further that there is extensive metabolic interaction between tumor cells and the surrounding microenvironment. More specifically, it has been observed that pathological cells are able to acquire highly energetic metabolites from the surrounding environment and utilize fatty acids from tissue adipocytes for energy production. It would also appear that glutamine and ketones produced by the stroma promote mitochondrial oxidative metabolism and energy production in cancer cells [14].

## 3. Ketogenic Diet Targets

### 3.1. Metabolic Targets

Cancer cells modify their metabolism to satisfy their energy, growth, and proliferation needs (Warburg effect) [16]. KD could have an antitumor effect through its action on both intra- and extracellular targets.

KD into the extracellular compartment causes a decrease in blood glucose; furthermore, evidence suggests that it reduces the levels of insulin and IGF-1, with consequent inhibition of the anabolic signals of the mTOR pathway [22,23].

On the intracellular side, ketone bodies, and in particular beta-hydroxybutyrate, have various consequences. Ketone bodies have a double effect as follows: beta-hydroxybutyrate is convertible to acetyl-CoA and so is able to enter inside the Krebs cycle, carrying out a positive action on healthy cells; meanwhile, in neoplastic cells, it acts as an obstacle to metabolism, as, due to the subversion of mitochondrial function, acetyl-CoA derived from it cannot generate ATP via the Krebs cycle; therefore, this acetyl-CoA could, on the other hand, be used by the neoplastic cell for lipogenesis and the synthesis of cholesterol [10].

Ketone bodies, from an intracellular side of the neoplastic cells’ point of view, can also act through their competitive activity on the monocarboxylate transporters as follows: this transporter is indeed used both for the entry of ketone bodies into the cell and for lactate export. The KD could therefore determine an increase in the intracellular concentration of lactate in the neoplastic cell, reducing the possibility of its exit, thus determining an effect on the growth and survival of cancer cells [24].

An additional metabolic intracellular target of KD in tumors is pyruvate kinase. Pyruvate kinase, of which there are four different isoforms, is the last enzyme of the glycolytic pathway [25]; furthermore, studies show that the M2 isoform is overexpressed in cancer cells by virtue of its pro-oncogenetic effect and the metabolic advantage it causes, mediated by the activation of HIF-1 [26]. In this context, in vitro studies have shown that KD would be able to inhibit the expression of this particular enzymatic isoform, thus affecting energy production and promoting apoptosis of glioblastoma cells. In this model, β-hydroxybutyrate would also lead to a significant reduction of other important enzymes such as hexokinase, lactate dehydrogenase, and pyruvate dehydrogenase, also reducing the expression of the GLUT-1 transporter [27].

Other important results of KD have been shown on glioblastoma mouse models, there are as follows: a reduction of HIF-1 alpha and VEGF receptor 2 expression (instead the expression of VEGF remains constant), reducing neoangiogenesis and limiting tumor metabolic changes [28].

KD was also shown to modify the expression of AQP-4 and zonula occludens-1 by reducing peritumoral edema [28]. 

### 3.2. Inflammation

Inflammation has recently been considered a cancer characteristic due to the increased local and systemic release of pro-inflammatory cytokines. Inflammation is indeed related to tumor genesis and progression through the hyperactivation of NF-kB and other transcription factors [29].

Fatty acids, by activating the PPAR-alpha receptor, lead to an inhibition of the NF-kB transduction pathway causing COX-2 and NOS-induced downregulation (overexpressed in tumors) [30].

Furthermore, KD, both alone and in association with supplementation of 6-diazo-5-oxo-1-norleucine (DON), a glutamine antagonist, has been shown to decrease the expression of TNF-alpha in glioblastoma models, thus reducing growth and inflammation and increasing long-term survival [31].

Another major player in the pro-inflammatory response in tumor pathology is the inflammasome, a large intracellular multiprotein signaling complex that plays a key role in the activation of inflammatory processes in response to pathogens or injuries, including cancer [32].

It has been observed that inhibiting the inflammasome reduces the growth of cancer, prolonging the survival of mouse models affected by glioma [33].

Beta-hydroxybutyrate can inhibit NLRP3 inflammasome assembly and NLRP3-mediated cytokine production, resulting in reduced inflammatory markers in CNS tumors [34].

From an immunotherapeutic side, in glioblastoma mouse models, KD has been shown to be able to activate anti-tumor immune responses. Specifically, it has been shown that KD is able to determine an increase in cytokines and cytolysis caused by CD8+ T lymphocytes, a greater infiltration of CD4+ T lymphocytes with the maintenance of normal levels of T reg lymphocytes, and a reduction in the expression of CD86 and PD-L1, thus reducing tumor-mediated immunosuppression [35].

KD has also been shown to be able to reduce peritumoral edema and steroid needs (see dedicated chapter) [28,36,37].

### 3.3. Oncogenes and Tumor Suppressors

Metalloproteases are a group of zinc-dependent endopeptidases capable of breaking down the extracellular matrix. Cancer cells overexpress these enzymes to promote local and systemic invasion. The ketogenic dietary regimen has been shown to reduce the expression of MMP-2 and MMP-9 (and of vimentin) [28]. 

The p53 is the main oncosuppressor that controls cell proliferation, apoptosis, and genetic stability. Physiologically, in healthy cells, p53 is functional and expressed at extremely low levels, while in neoplastic cells p53 is characteristically mutated and overexpressed, often conferring resistance to therapy. KD would be able to determine a downregulation of mutated p53 through deacetylation with induction of death in neoplastic cells [38,39,40,41].

AMP kinase is an enzyme that can activate tumor suppressors such as p53, suppressing growth and arresting the cell cycle. So far, several molecules have been shown to be able to activate AMPK, including metformin, curcumin, some NSAIDs, and even ketone bodies [41,42].

Comparing the use of the KD and the standard diet on animal tumors, it emerged that KD would be able to decrease the expression of pathways not only mediated by IGF-1, but also by PDGFR and EGF, pathways frequently overexpressed in gliomas and leading to activation of Akt and mTOR. mTOR is responsible for the activation of transcription factors such as HIF-1, which increase the transcription of oncogenes, GLUT, and glycolytic enzymes [43,44]. Indeed, KD enhances tumor response to PI3K inhibitors [45].

### 3.4. ROS

To support their growth, cancer cells also develop alterations at the mitochondrial level, resulting in an increase in ROS production [27].

ROS production and oxidative stress, however, are a double-edged sword for neoplastic cells as follows: on one hand, they confer an “evolutionary” advantage linked to the higher mutation rate with the creation of new mutations and increasingly diversified clones; on the other hand, there is the concrete risk of overcoming a critical oxidative stress value, which would cause too much tumor damage that cannot be repaired by neoplastic cells [46]. This is how conventional therapies such as radio- and classical chemotherapy work as follows: they try to make the damage to cancer cells go beyond this irreversibility threshold, in order to cause their death [47].

KD, by reducing the availability of gluocose-6-phosphate, prevents the correct functioning not only of glycolysis but also of the pentose phosphate shunt, an alternative metabolic pathway that is overexploited by neoplastic cells as it is able to supply reduced NADP, an essential component for recharging the reduced glutathione and allows the control of ROS levels and cellular oxidative stress.

However, this path also allows the conversion of sugars with 6 carbon atoms such as glucose into sugars with 5 carbon atoms such as ribose, an essential component of nucleic acids, thus also determining a reduction in tumor growth and proliferation [48]. 

The pro-oxidative effect, on the other hand, does not occur in healthy tissues, which, on the contrary, would be protected by the ketogenic diet since beta-hydroxybutyrate would be able to trigger a mitochondrial uncoupling protein (UCP-2) and improve the cellular antioxidant response [49].

The overall positive effect would be twofold, as follows: on the one hand, the toxic synergistic effect of KD with conventional therapies on cancer cells; on the other hand, the antioxidant and protective effect of KD on healthy tissues [43,50].

### 3.5. Epigenetic Modulation

The effect of KD on the genome and its expression is new and is still largely a topic yet to be studied. KD could be able to modulate gene expression both directly, through the regulation of DNA methylation (KD increases adenosine causing a block of DNA methylation [51]), and indirectly, through the modification of the histone conformation, through acetylation, methylation, phosphorylation, ubiquitylation and lysine beta-hydroxybutyrylation; this last histone transformation would seem exclusive of ketone bodies. These epigenetic modifications would explain the ability of KD to positively modify the expression of oncogenes and tumor suppressors [10,41,52].

An even lesser-known topic of great scientific interest is that of microRNAs (miRNAs) as follows: these molecules can modify gene expression by linking sequences of complementary mRNA, directing them to degradation, resulting in the silencing of some genes. MiRNAs are implicated in several pathological conditions including cancer; in all the tumors analyzed, indeed, altered miRNA expressions were found resulting in up-regulation of oncogenes and down-regulation of tumor suppressors [53].

Changes in miRNA expression have also been demonstrated in glioblastomas, in particular, the following: 256 significantly overexpressed miRNA (mainly miR-10b, miR-17-92 clusters, miR-21, and miR-93) and 95 significantly under-expressed miRNAs (in particular miR-7, miR-34a, miR-128, and miR-137) in the GBM compared to healthy brain tissue [54].

In glioblastoma mouse models, KD has been shown to be able to modulate the expression of various miRNAs, reducing tumor progression and increasing long-term survival [41,55,56] (Figure 4).

## 4. Pre-Clinical Studies

Early studies, conducted in vivo in animal models about the possible use of diet in brain tumor therapy, hypothesized the combined intervention of different dietary approaches as follows: fasting, intermittent fasting, caloric restriction, and/or ketogenic diet (with or without caloric restriction). These approaches have most often proved successful, reducing tumor growth, and increasing survival [57,58].

Some preclinical studies conducted in glioblastoma mouse models involving exclusive treatment with 3:1 KD showed no positive effects on tumor progression and survival [59,60]; whereas, when the ratio of fatty acids to carbohydrates was increased to 4:1 or 6:1, an effect on survival was observed as determined by both KD alone [52,61] and in combination with RT, with a synergistic effect of the two approaches [62]. A similar result is also stressed for the association between KD and bevacizumab; indeed, the two approaches in combination seem to have a synergistic effect as follows: they increase survival and reduce tumor volume and ATP concentration [60]. Another study also showed that KD both alone and in combination with RT and temozolomide is able to prolong survival and slow tumor growth in mouse models of glioblastoma compared with those receiving the same treatment but maintained on a standard diet (SD) [43].

In contrast, when mouse models of medulloblastoma are considered, even increasing the ratio of fatty acids to carbohydrates to 4:1 and then to 6:1 does not show any benefit in terms of survival or tumor progression [63].

In two other studies on mouse models of two different glioblastoma lines, different dietary combinations were considered, using SD, low-calorie SD, KD, and low-calorie KD in one study and SD, SD + DON, KD, and KD + DON in the other. The results indicate that both KD and caloric restriction can significantly reduce tumor growth and prolong survival; moreover, the combination of KD + DON showed a synergistic effect compared with SD + DON and KD approaches taken individually, resulting in a positive effect on survival, tumor growth, inflammation, and peri-tumor edema [31,64].

A more recent pre-clinical study conducted in mouse models with high-grade glioma showed that KD compared with SD was more effective in improving survival. This study also evaluated the difference in some metabolites between healthy and tumor brain tissue in mice treated with SD and KD. There is evidence of the following: (a) lower concentration in the neoplastic tissue of KD mice of amino acids critical for tumor cells such as glycine, N-acetylaspartate, and N-acetylaspartyl-glutamate, with instead normal levels of these amino acids in healthy brain tissue; (b) markedly higher concentrations of beta-hydroxybutyrate in the tumor tissue of KD mice, demonstrating the inability of these tumors to utilize ketone bodies for energy production, with minimal levels instead of BHB in healthy tissue, which is capable of utilizing this molecule in mitochondrial activity; (c) finally, cellular energy photography shows that in the tumor tissue of KD mice there is a decrease in the concentration of creatine and phosphocreatine levels, a decrease that does not occur in healthy tissue [65].

However, the results of KD are still discordant, as follows: in a 2016 study using KD in mice with glioma, it was observed that rat gliomas are able to oxidize ketone bodies and overexpress monocarboxylate transporter 1 (MCT1) when nourished with a ketogenic diet. These results would contradict the hypothesis that brain tumors are metabolically rigid and show the need for further research on the use of ketogenic diets as a therapy targeting brain tumor metabolism [66].

There have been several meta-analyses of pre-clinical studies of tumors in animal models treated with KD. They show a benefit of KD on survival; however, these studies do not exclusively consider brain tumors [58,67].

Regarding specific mutations and their response to KD in the literature, there are not many studies yet. In recent years, the mutation of isocitrate dehydrogenase, an enzyme that is part of the Krebs cycle and frequently mutated in gliomas, has been studied in particular. From the studies, there seems to be no difference in the use of KD on wild-type and IDH mutated IDH gliomas [68].

The main pre-clinical studies in mouse models are summarized in Table 1.

## 5. Clinical Studies

Clinical studies on the use of KD in CNS tumors, especially in the context of treating gliomas, have increased in recent years; the main evidence observed from the literature review will be reported below, considering that the sample size of these studies still remains very limited.

In 2014, in the ERGO study, 20 patients with recurrent glioblastoma treated with KD were analyzed; furthermore, patients in this study were treated with KD after a conventional therapeutic approach followed by any salvage therapy in case of progression. In this study, it was observed that all patients showed disease progression, demonstrating that KD is not sufficient as an exclusive treatment for this type of disease. However, a longer trend in progression-free survival (PFS) was observed in subjects with stable ketosis compared with patients with unstable ketosis. Furthermore, patients treated with bevacizumab + KD had a PFS of approximately 20.1 weeks, compared with 16.1 weeks for patients from the same center treated with bevacizumab monotherapy [60].

A retrospective study of 53 patients with grade III and IV glioma treated with standard therapies was also conducted in 2014. Of these, 6 subjects simultaneously followed a KD, demonstrating the safety and good tolerability of KD combined with CT and/or RT. Glucose restriction obtained through KD appeared to significantly reduce serum glucose levels, also in patients simultaneously taking high-dose steroids, possibly improving response to standard treatment and prognosis [69].

In relation to patients’ quality of life, a 2020 study of different types of gliomas demonstrated the efficacy of KD. In this analysis, 12 patients treated with standard therapy and KD 3:1 were considered; they saw improvement in energy, mood, neuro-cognitive function, and general well-being, as well as decreased seizures. In addition, in 4 patients, imaging illustrated a reduction in contrast enhancement or T2 hyperintensity, suggesting a possible neuroradiological response of diet combined with standard therapy [70]. Another study also applied a particular type of KD, the Modified Atkins diet (MAD), combined with standard therapy to gliomas of different grades, demonstrating the ability of this approach to better control seizures and improve patients’ quality of life [44]. Even more recent is the 2020 KEATING study, where 12 patients with glioblastoma treated with standard RT and CT combined with MAD or MCT diet were studied. Only 4 patients completed the 3-month diet, however, confirming the positive effect on GHS of patients following MAD [71].

In contrast, a 2019 study of 11 glioblastoma patients treated with conventional therapy combined with KD 4:1, although it showed tolerability of the diet, showed no changes in quality of life, neurological functioning, or the survival of the 6 patients who completed the study [72].

The ketogenic approach combined with standard therapy was also tested in three children with diffuse intrinsic pontine glioma; in this study, the dietary approach was shown to be safe, but because the sample size was very small, no significant data emerged in terms of increased survival [73]. These results were also later confirmed by a more recent review of diffuse intrinsic pontine glioma in children [74].

Finally, a meta-analysis on the use of KD combined with standard therapies in gliomas was also recently conducted. In most of the studies, overall survival exceeded the average prognosis of patients receiving standard therapy, although the small sample size, the presence of different types of glioma, and the absence of a control population preclude drawing solid results about the actual effectiveness of KD in treating gliomas [3].

There are still many active trials trying to evaluate the impact of KD on gliomas [75].

The main clinical studies on the use of KD in CNS tumors are summarized in Table 2.

## 6. Ketogenic Diet and Steroids

The standard of care for CNS tumors may include the use of dexamethasone for short periods to reduce peritumoral edema and intracranial pressure. The ketogenic diet and steroid use are not mutually exclusive treatment approaches, and they can also be used in combination. From the few data in the literature, there is evidence that KD is able to reduce the need for high doses of dexamethasone, to reduce peritumoral edema, going to better delineate tumor margins [28,36,37]. In addition, in patients treated with KD and steroids, lower than expected blood glucose elevation was noted as follows: in this context, KD could also be useful in the prevention of hyperglycemia induced by steroid therapy [69].

## 7. The Anti-Epileptic Effect of the Ketogenic Diet

Gliomas are highly epileptogenic tumors; indeed, the most typical onset symptom of these cancers is a seizure, which occurs at an incidence of 65 to 90% [80]. The ketogenic diet could also play a role in controlling the symptomatology of such tumors by virtue of its anti-epileptic action, which has already been illustrated in the literature. Indeed, it is known from various scientific evidences that KD has an anti-epileptic effect on drug-resistant epilepsy, especially in children [81]. The mechanisms underlying this anticonvulsant effect are not completely known, but it is hypothesized that ketone bodies and polyunsaturated fatty acids, produced by KD, are responsible for it [82]. Indeed, the underlying rationale predicts that these two molecular classes, and KD more generally, may result in the following at the neuronal level: (a) increased ATP and phosphocreatine production capacity, with increased resistance to metabolic stress and increased seizure threshold; (b) the increase in the opening of ATP-sensitive potassium channels, resulting in hyperpolarization of the neuronal membrane; this is associated with the regulation of membrane excitability mediated again by ketone bodies and fatty acids, due to the activation of the two-pore domain of potassium channels; (c) the activation of glutamic acid decarboxylase, with synthesis of GABA, as well as the possible alteration of the transaminase activity of GABA, with inhibition of its degradation; (d) the increased levels of agmantin, an inhibitory neurotransmitter, at the hippocampal level; (e) the alteration of serotonin and dopamine levels (important in the control of neuronal excitability) in the cerebrospinal fluid; (f) the upregulation of calbindin, with neuroprotective potential by virtue of its ability to buffer intracellular calcium, and by inhibition of apoptotic factors such as caspase 3 (reducing excitotoxicity and apoptosis, the main mechanisms involved in neuronal damage and death related to severe and long-lasting seizures) [83]. 

## 8. Positive Effects, Negative Effects, Side Effects, and Doubts of KD

Evidence suggests that KD could be a valuable weapon to be used synergistically with conventional therapies in the fight against tumors, particularly those of the CNS. Its effect in this context does not appear to relate exclusively to the reduction of tumor growth and mass in mouse models [43,50], but also seems to be associated with the following: (a) a protective action for healthy cells against CT and RT [43,50], (b) increased survival in mice with gliomas, [31,43,64]. (c) reduction of tumor standardized uptake values (SUV) in pediatric patients [84], (d) reduction of seizures [83,85], (e) reduction of steroid requirement [28,36,37], (f) reduction of peritumoral edema [28,36,37], (g) reduction of neoplastic infiltration with facilitation of surgery [28,36,37], and (h) improvement of sleep and mood [86].

In contrast, there are studies hypothesizing a possible adaptation of glioma to the ketogenic environment, which could result in a lack of efficacy of dietary treatment [66]. In addition, the “reverse Warburg effect” has also been theorized from studies of breast tumors as follows: this theory hypothesizes a possible utilization of ketone bodies, produced by peritumor fibroblasts, within the Krebs cycle of neoplastic cells [87].

It should be emphasized that the possible use of KD as cancer therapy cannot be considered as a cross-cutting intervention as follows: in addition to the general contraindications well summarized and explained by Watanabe et al. [88]. There are also specific contraindications concerning certain tumors; indeed, there is evidence in mouse models about increased growth, when treated with KD, of renal tumors associated with tuberous sclerosis and tumors with BRAF mutation V600E [89,90].

KD is also not without side effects, as follows: among those most frequently experienced are fatigue, muscle cramps, hypotension, constipation, and unwanted weight loss [8]. In most of the studies analyzed in this review, the application of KD was shown to be safe with the occurrence of only minor adverse effects such as constipation, diarrhea, nausea/vomiting, hypercholesterolemia, hypoglycemia at the beginning of the diet, fatigue, alopecia, nephrolithiasis, dizziness, headache, hyperuricemia, reduction in carnitine levels, anorexia, and refusal to feed. These effects, however, were usually transient and often also related to standard therapies associated with the diet [3,60,72,74]. The presence of these possible side effects could at least partly explain the significant KD dropout rate, which was consistently present in almost all the studies analyzed [60,71,72]. Serious issues emerged in the following two articles: in one study, hydroelectrolyte disorders are described [71] and in the other, the occurrence of DVT in a patient presenting MTHFR mutation [69]. 

Studies in the literature on the application of KD in cancer disease show contrasting results about the improvement in quality of life, and doubts still remain about the ability of KD to improve both cachexia and fatigue [23,41].

## 9. Conclusions

Most of the pre-clinical and clinical data analyzed in this review suggest that the ketogenic diet can play a positive role in therapy against CNS tumors by contrasting their metabolism, inflammation, pathological gene transcription, and tumor microenvironment. Positive factors certainly include its limited toxicity, low cost, and easy application. Difficulties, on the other hand, include possible side effects as well as poor compliance, leading to significant dropout from studies. In addition, an issue not yet considered is the possible metabolic plasticity of cancer cells as follows: they may be able to escape glucose dependence on other metabolic pathways through metabolic reprogramming [14].

In the near future, the ketogenic diet is likely to be proposed as a combination therapy with conventional CT and RT because of its possible dual benefits as follows: on the one hand, the synergistic toxic effect on cancer cells; on the other hand, the possible protection of healthy cells from the toxicity of standard therapies, given its ability to increase cellular oxidative capacity. In addition, there may be a possibility of using KD to control CNS tumor-induced seizures because of the known antiepileptic activity of this diet. 

However, it is necessary to improve clinical trials by testing KD on a larger scale to obtain more robust data that may or may not confirm the results of using this diet in CNS tumors. To do this, it would be important for every oncology department to have nutrition experts working and collaborating with the medical staff to integrate the classical therapeutic approach with the nutritional one. It is also necessary to structure studies, both in vivo and in vitro, designed to clarify definitively the response of various cells (both healthy and nonhealthy) to the ketogenic environment, ultimately uncovering all the mechanisms underlying the therapeutic implications of the ketogenic diet.

## Figures and Tables

**Figure 1 nutrients-14-03851-f001:**
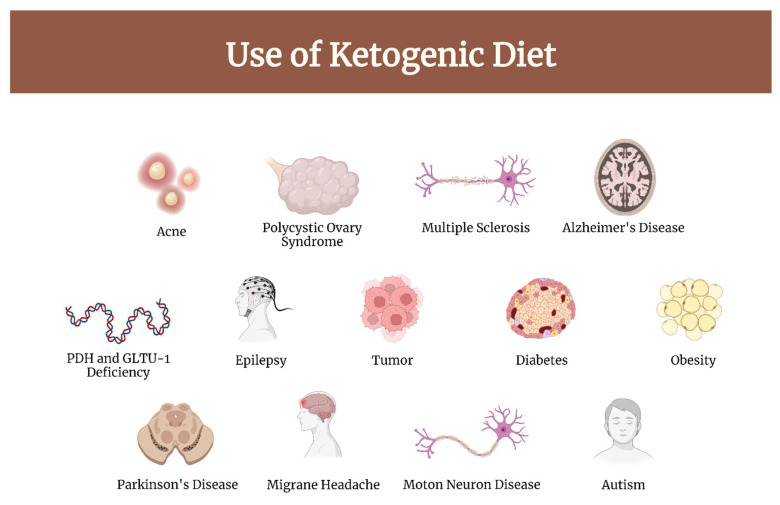
Diseases in which KD has demonstrated clinical efficacy.

**Figure 2 nutrients-14-03851-f002:**
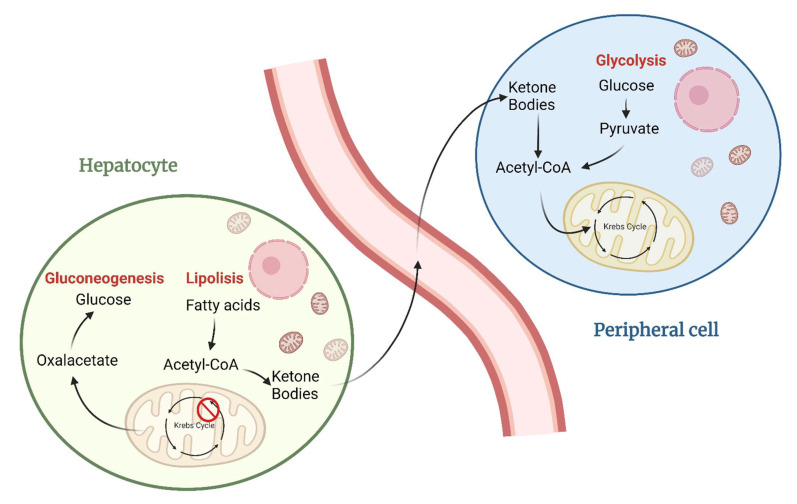
Mechanism of action in the production and utilization of ketone bodies.

**Figure 3 nutrients-14-03851-f003:**
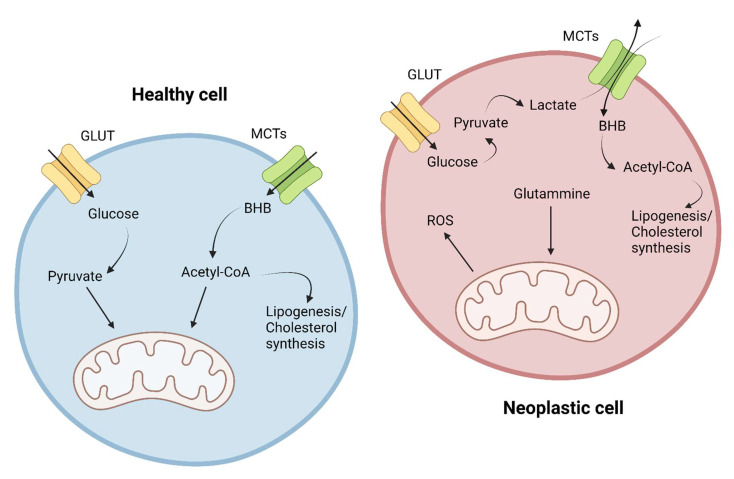
Metabolic differences in the utilization of glucose and ketone bodies in healthy cells and cancer cells.

**Figure 4 nutrients-14-03851-f004:**
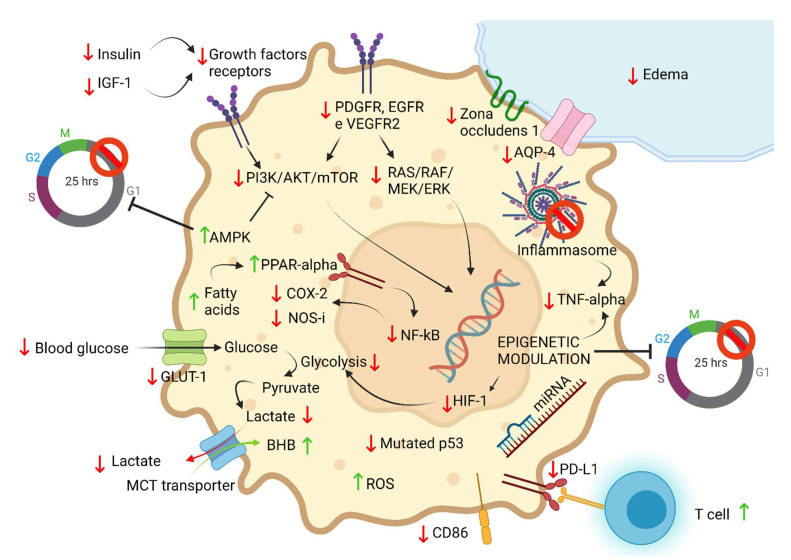
Main mechanisms of functioning of ketogenic diet against cancer cell described in the literature (↑ = increased; ↓ = reduced).

**Table 1 nutrients-14-03851-t001:** The main pre-clinical studies in mouse models.

Cancer Type	Year	Cell Line	KD Ratio	Study Group	Results	Reference
Glioma	2003	CT-2A	5.5:1	SD, KD, SD + CR, KD + CR	Reduced tumor growth in groups with CR.	[57]
	2007	CT-2A, U87-MG	4:1	KD + CR vs. SD + CR	KD reduced growth and improved survival.	[61]
	2010	GL261	6:1	SD vs. KD	KD increased survival.	[52]
	2011	T98G, U87MG, NIH3T3, A172, LNT-229, U251MG	3:1	SD vs. KD	No effects on survival or tumor progression.	[59]
	2012	GL261-Luc2	4:1	KD, SD + RT, KD + RT	KD alone and KD + RT increase survival with an additive effect compared with RT alone.	[62]
	2014	U87MG	3:1	SD, SD + bevacizumab, KD, KD + bevacizumab	KD alone not effective on tumor progression or survival. KD + bevacizumab: synergistic effect.	[60]
	2015	GL261-Luc2	4:1	SD vs. KD	In KD the expression of VEGFR2, MMP-2, MMP-9, vimetina and peritumoral edema were reduced.	[28]
	2015	SH-SY5Y, SK-N-BE(2)	1.5:1	SD, CR, KD, KD + CR	Ketogenic diet and/or calorie restriction significantly reduced tumor growth and prolonged survival.	[64]
	2016	L9, RG2	4.5:1	SD vs. KD	Gliomas can oxidize ketone bodies and overexpress Monocarboxylate transporter 1 (MCT1).	[66]
	2019	VM-M3, CT-2A	3.6:1	SD, SD + DON, KD, KD + DON	KD + DON increased the survival, reduced the tumor growth and edema.	[31]
	2020	GL261	7:1	SD vs. KD	Improved survival and changes in the availability of energy and structural resources of neoplastic cells.	[65]
	2022	GBM6, GBM43, TB09 e HT1080	4.2:1	KD vs. SD	There are no differences in the use of KD in wild-type and mutated IDH.	[68]
Medulloblastoma	2015	Spontaneous medulloblastoma (*Ptch1^+/−^Trp53^−/−^*)	4:1 and 6:1	SD vs. KD	No effects on the tumor progression or survival.	[63]

**Table 2 nutrients-14-03851-t002:** The main clinical studies on the use of KD in CNS tumors.

Cancer Type	Year	Number of Patients	Study Group	Results	Reference
Recurrent glioblastoma	2014	20	KD, conventional therapy, KD + conventional therapy	KD alone not efficacy. KD + bevacizumab prolongs PFS compared with bevacizumab alone.	[60]
Glioblastoma	2014	6	Standard therapy vs. terapia standard + KD	KD well tolerated and safe even in combination with standard therapies. Improved glucose profile also in combination with steroids.	[69]
Glioma	2015	8	Standard therapy + MAD (modified atkins diet)	KD well tolerated with improved seizure control.	[44]
Glioblastoma and gliomatosis cerebri	2017	9	SD, KD, KD+bevacizumab	KD determines accumulation of ketone bodies in the CNS of patients with brain tumors.	[76]
High-grade glioma	2018	6	Standard therapy + MKD	Well-tolerated diet with limited side effects (fatigue, constipation).	[77]
Glioblastoma	2019	11	Standard therapy + KD	No severe adverse effects, no effects on survival, neurological functioning, or quality of life.	[72]
Difuse intrinsic pontine glioma	2019	3	Standard therapy + KD	KD is safe but the effect on survival requires a larger cohort.	[73]
Glioblastoma	2020	8	Standard therapy + KD	KD was well tolerated, sample sparsity did not allow testing for survival benefits.	[78]
Glioblastoma	2020	12	Standard therapy+MKD (modified ketogenic diet) o MCTKD (medium-chain triglycerideketogenic diet)	Some patients developed indroelectrolyte disorders. There was an improvement in GHS, which was better in MKD.	[71]
Glioma	2020	12	Standard therapy + KD	Improved symptoms and seizures. Improved disease control with reduction in vasogenic edema.	[70]
High-grade glioma	2021	13	RT + modified atkins diet + MCT + metformin supplementation	Promising intervention.	[79]
Diffuse intrinsic pontine glioma	2021	5	Standard therapy + KD	KD is safe but the effect on survival requires a larger cohort.	[74]

## Data Availability

Not applicable.

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
