# Peer review of "Ketogenic Diet in the Treatment of Gliomas and Glioblastomas"

_nutrients, 2022, doi:10.3390/nu14183851_

Round 1

Reviewer 1 Report

Well written comprehensive review of ketogenic diet and potential use in treatment of gliomas .

Only addition I would suggest is to give more opinion or recommendations on what is lacking in the research endeavours both in basic  science and clinical research. In other words the limitations of this review..

Author Response

Thank you for the review and suggestion. I added a short paragraph in the conclusion that talks about what is lacking in research efforts in both preclinical and clinical settings and the problem of limited clinical data available.

Reviewer 2 Report

In the manuscript” Ketogenic diet in the treatment of gliomas and glioblastomas”, the authors summarized the current progression of KD on metabolic, inflammatory, oncogenic and oncosuppressive, ROS and epigenetic modulation. And they also reviewed pre-clinical studies and clinical studies. The manuscript is well written and presented in a logical way. The following issues should be addressed.

1. In 3.3, “AMP kinase is an enzyme capable of activating tumor suppressors such as p53 and LKB1, suppressing growth and stopping the cell cycle.”, AMPK is the downstream of LKB1 and activated by LKB1.

2. The legend of Figure 4 needs to be revised.

3. The authors need to review the manuscript carefully. There are some typos and tiny mistakes, such as in 3.3 “p53 is the main oncosoppressor which controls cell proliferation” oncosuppressor, not oncosoppressor; in 3.1 “Other important results of KD have shown on glioblastoma  mouse models,” it seems that there are two spaces between the words glioblastoma and mouse.

Author Response

Thank you for the review and very helpful comments to improve the manuscript.

  1. AMPK is downstream of LKB1 and activated by LKB1, so I rephrased the sentence removing the error.
  2. I revised the figure 4 legend that had been incorrectly edited.
  3. I revised the entire manuscript by correcting typos and tiny mistakes.